# Diagnostic Accuracy of Magnetic Resonance Imaging in the Detection of Type and Location of Meniscus Tears: Comparison with Arthroscopic Findings

**DOI:** 10.3390/jcm10040606

**Published:** 2021-02-05

**Authors:** Seong Hwan Kim, Han-Jun Lee, Ye-Hoon Jang, Kwang-Jin Chun, Yong-Beom Park

**Affiliations:** 1Department of Orthopedic Surgery, Chung-Ang University Hospital, Chung-Ang University College of Medicine, 102 Heukseok-ro, Dongjak-gu, Seoul 06973, Korea; ksh170177@nate.com (S.H.K.); gustinolhj@cau.ac.kr (H.-J.L.); jyh1029@caumc.or.kr (Y.-H.J.); ckj937@caumc.or.kr (K.-J.C.); 2Department of Orthopedic Surgery, Hyundae General Hospital, Chung-Ang University, 21 Bonghyun-ro, Jinjup-Eop, Namyangju-si 12013, Korea

**Keywords:** meniscus tear, magnetic resonance image, accuracy

## Abstract

Magnetic resonance imaging (MRI) has been widely used for the diagnosis of meniscal tears, but its diagnostic accuracy, depending on the type and location, has not been well investigated. We aimed to evaluate the diagnostic accuracy of MRI by comparing MRI and arthroscopic findings. Preoperative 3.0-T MRI and arthroscopic findings from 2005 to 2018 were reviewed to determine the presence, type, and location of meniscus tears. In addition, subgroup analysis was performed according to anterior cruciate ligament (ACL) injury. The exclusion criteria were as follows: (1) Inflammatory arthritis, (2) other ligament injuries, (3) inability to classify meniscal tears due to degenerative arthritis, (4) over 90 days from MRI to surgery, and (5) incomplete data. Of the 2998 eligible patients, 544 were finally included. The sensitivity and specificity of MRI in determining medial and lateral meniscus tears were 91.8% and 79.9%, and 80.8% and 85.4%, respectively. The accuracy of MRI in the ACL-injured group was lower than that in the ACL-intact group (medial meniscus: 81.7% vs. 88.1%, *p* = 0.041; 72.9% vs. lateral meniscus: 88.0%, *p* < 0.001). MRI accuracy was low for the longitudinal tears of the posterior horn of the medial meniscus in the ACL-injured group. MRI could be a diagnostic tool for meniscus tears, but has limited accuracy in their classification of the type and location. Hence, care should be taken during arthroscopic assessment of ACL-injured patients due to low diagnostic accuracy of preoperative MRI.

## 1. Introduction

Meniscus tears are clinically important due to the role of the meniscus in load distribution, shock absorption, and joint stabilization [1,2]. Meniscus root tear or meniscectomy increases joint contact pressure [3], which can lead to articular cartilage degeneration over time [4,5,6]. The treatment of meniscal tears needs to be determined according to the location and type of meniscal tears. Therefore, correct identification of a meniscal tear is crucial to its subsequent management.

Magnetic resonance imaging (MRI) has been commonly used to detect meniscal tears in clinical practice. Despite the evolution of MRI in detecting meniscal tears, several lesions in meniscal tears are missed, given the wide variability in the accuracy of MRI [7,8,9,10,11]. In particular, the sensitivity of MRI in the detection of ramp lesions, meniscocapsular separation, or peripheral longitudinal tear of the posterior horn of the medial meniscus in the anterior cruciate ligament (ACL)-injured knee was relatively low, approximately 65.1% [11,12]. In addition, a high rate (40–67.7%) of missed lateral meniscus posterior root tears on MRI has been reported [8,13]. However, previous studies evaluated a relatively small number of patients who underwent ACL reconstruction [8,12]. In addition, very few studies have evaluated the accuracy of MRI in the detection of meniscal tears according to type and location.

Therefore, we sought to evaluate the diagnostic accuracy of MRI in detecting meniscal tears according to type and location by comparing MRI and arthroscopic findings. In addition, the accuracy of MRI was compared based on the presence or absence of ACL injury.

## 2. Materials and Methods

This was a retrospective cohort study of 2998 patients who had undergone arthroscopic surgery between November 2005 and August 2018. The indications for arthroscopy were medial or lateral meniscal tears with/without ACL rupture on MRI. Confirmative diagnosis of meniscal lesions were made using the arthroscopic findings according to the location and classification of the International Society of Arthroscopy, Knee Surgery, and Orthopedic Sports Medicine (ISAKOS) [14]. The ACL rupture was assessed by MRI findings, clinical examination, stress radiography and associated symptoms and functional disability. The exclusion criteria were as follows: (1) Inflammatory arthritis, such as gout, infection, rheumatoid arthritis, etc. (177 patients), (2) previous history of surgery on the knee or other ligament injuries (601 patients), (3) inability to classify meniscal tears due to degenerative arthritis, (288 patients), (4) over 90 days from MRI to surgery (371 patients), and (5) incomplete clinical data or arthroscopic findings (1018 patients with insufficient information about the type and location of meniscus tears). After applying all these criteria, 543 patients (379 male and 164 female) were finally included in this study (Figure 1). The study was approved by the Institutional Review Board at our hospital and was performed in accordance with the ethical standards of the 1964 Declaration of Helsinki.

### 2.1. MRI Evaluation

MRI examinations were performed on a 3.0-T scanner (Achieva, Philips, Amsterdam, Netherlands) with an eight-channel knee coil (HD time of repetition [TR] knee array). Coronal and sagittal images were acquired using turbo spin-echo (TSE) proton density-weighted imaging (PDWI), TSE fat-saturated T2-weighted imaging (T2WI), T1-weighted sequence, axial imaging using TSE PDWI, and ACL oblique imaging using TSE fat-saturated T2WI. The TR range and time of echo values varied (1500–3000/10–30 ms for PDWI and 5000–6000/100 ms for T2WI). Other imaging parameters used were as follows: Field of view, 16 cm; slice thickness, 1.5 mm with no space. The minimal tear of meniscus was defined using the “two-slice-touch” rule [15].

The location and type of meniscal tear were evaluated using the ISAKOS classification [14]. The locations of meniscal tears were classified as follows: (1) Anterior horn, (2) mid-body, (3) posterior horn, and (4) involvement of more than two compartments. The types of meniscal tears were categorized as follows: (1) Longitudinal-vertical tear including bucket handle tear, (2) horizontal tear, (3) radial tear, (4) vertical flap tear, (5) horizontal flap tear, and (6) complex tear. The root tears of meniscus were classified as complex tear, due to no category for root tear in ISAKOS classification. Only the grade III a horizontal tear on MRI was included as horizontal tear on MRI.

### 2.2. Arthroscopic Evaluation

All surgical procedures were performed using the same technique in our institution [16,17,18,19]. Systematic arthroscopic evaluations were performed using 30° and 70° arthroscopes with/without ACL reconstruction. First, standard anterolateral portals were used for routine evaluation; then, the posteromedial and posterolateral compartments of the knee were visualized through the intercondylar notch using 30° and 70° arthroscopes [16,17]. The type and location of meniscal tears were assessed by probing in accordance with the ISAKOS classification [14]. After systematic evaluations, the meniscus tears were treated (repair or meniscectomy) with or without ACL reconstruction. Data of the type and location of meniscal tears were collected from arthroscopic surgical report form referring to arthroscopic images.

### 2.3. Statistical Analysis

The results were analyzed using statistical software (SPSS 19.0; Chicago, IL, USA), and power analyses were performed using the G*power program (ver. 3.1.5). To compare the values between groups, the data were analyzed by the Mann-Whitney *U*-test, independent t-test, paired t-test, chi-squared test, or Wilcoxon signed-rank test according to the results of the Shapiro-Wilk test used to test the normality of their distribution. Arthroscopic findings were used as the reference standard. The sensitivity, specificity, positive predicted value (PPV), and negative predicted value (NPV) were calculated and compared between groups using the Cochran-Mantel-Haenszel test according to the ISAKOS classification and location of meniscus tear. Subgroup analysis was performed according to the presence of ACL injury with instability more than grade 2.

Post-hoc power analysis was performed using a χ^2^ goodness-of-fit test with an α error of 5% to detect the differences in the incidence of type and location of tear between groups. Based on these calculations, the power of all the significant variables in this study was over 0.95, indicating robustness. Statistical significance was set at *p* < 0.05. All radiographic or MRI studies were reviewed by two musculoskeletal radiologists, who had more than 3 years of experience, with an interval of 2 weeks. The inter- and intra-observer reliability of the measurements was assessed using the kappa value for agreement or the intraclass correlation coefficient (ICC) for consistency.

## 3. Results

Of the 543 patients, 192 (35.4%) were had confirmed ACL injury on arthroscopy. In total, there were 261 cases of medial meniscus tear (48.1%) and 271 cases of lateral meniscus tear (49.9%) on arthroscopy. However, on MRI, there were 302 cases of medial meniscus tear (55.6%) and 258 cases of lateral meniscus tear (47.5%). The details of the patients’ demographic data are summarized in Table 1. The overall type and location of meniscal tears are shown in Figure 2 and Figure 3.

### 3.1. Diagnostic Accuracy of MRI for Meniscus Tears

The sensitivity and specificity of preoperative MRI in determining medial meniscus tears were 91.82% and 79.93%, respectively, with an accuracy of 85.8%. The PPV and NPV were 81.79% and 90.87%, respectively, indicating that preoperative MRI had better sensitivity and NPV in the diagnosis of medial meniscus tears than specificity or PPV. The kappa value between MRI and arthroscopic findings was 0.717, indicating good agreement (*p* = 0.03). The sensitivity and specificity of preoperative MRI in determining lateral meniscus tear were 80.74% and 85.35% with 83.1% accuracy. The PPV and NPV were 84.49% and 81.75%, respectively, indicating that preoperative MRI had better specificity and PPV in the diagnosis of lateral meniscus tear than sensitivity or NPV. The kappa value between MRI and arthroscopic findings was 0.661, indicating moderate agreement (*p* = 0.032). The sensitivity and NPV of preoperative MRI were higher in the diagnosis of medial meniscus tear than in diagnosis of lateral meniscus tear, but the specificity and PPV were higher in the diagnosis of the latter than in the diagnosis of the former.

### 3.2. Location and Type of Meniscus Tears in ACL-Injured Patients

The overall sensitivity and specificity of MRI in diagnosing medial meniscus tears were found to be 87.9% and 75.3%, respectively, with an accuracy of 81.7%. The PPV and NPV were 79.1% and 85.4%, respectively. However, the sensitivity of detecting all types of medial meniscus tears was 33.3% to 76.5%, which might be fair to moderate, although the specificity and accuracy were found to be 79.2% and 99.5%, respectively (kappa value = 0.511, *p* = 0.045; Table 2 and Appendix A). The accuracy of MRI in determining the location of medial meniscus tear was found to range from 78.1% to 98.4%. (kappa value = 0.519, *p* = 0.048; Table 2 and Appendix A).

The overall diagnostic values of MRI in determining lateral meniscal tears were 67.0% sensitivity, 79.4% specificity, 77.9% PPV, 68.9% NPV, and 72.9% accuracy. The sensitivity of the complex tear was as high as 89.9% (kappa value = 0.408, *p* = 0.048; Table 3 and Appendix A). The accuracy of MRI in determining the location of the lateral meniscus tear was 76.6–96.4%. (kappa = 0.396, *p* = 0.05; Table 3 and Appendix A).

### 3.3. Location and Type of Meniscus Tears in ACL-Intact Patients

The overall diagnostic values of medial meniscal tears on MRI were 96.3% sensitivity, 80.9% specificity, 81.3% PPV, 96.2% NPV, and 88.1% accuracy. However, the sensitivity of MRI in identifying all types of medial meniscus tears was 38.5% to 69.2%, which is fair–moderate, although the specificity and accuracy were 83.5% and 98.2%, respectively (kappa value = 0.573, *p* = 0.033; Table 4 and Appendix A). The overall accuracy of MRI in determining the location of medial meniscus tears was 82.6–98.3% (kappa value = 0.548, *p* = 0.033; Table 4 and Appendix A).

The overall diagnostic values of MRI in determining lateral meniscal tears were 94.1% sensitivity, 82.3% specificity, 83.3% PPV, 93.7% NPV, and 88.0% accuracy. However, the sensitivity of all types of lateral meniscus tears was 44.8–75.0%, which is fair–moderate, while the specificity and accuracy were found as 83.5% to 99.7% (kappa value = 0.568, *p* = 0.032; Table 5 and Appendix A). The overall accuracy of MRI in determining the location of lateral meniscus tears was 87.5–94.0% (kappa = 0.582, *p* = 0.032; Table 5 and Appendix A).

### 3.4. Comparison of the Type and Location of Meniscus Tears between ACL-Injured and ACL-Intact Patients

For the medial meniscal tears, the accuracy of MRI in patients with ACL injury was found to be lower than that in ACL-intact patients (81.7% vs. 88.1%, *p* = 0.041). There were differences in the rate of tear type (*p* < 0.001), but not in the location of tears between ACL-injured and ACL-intact patients (*p* = 0.254). In the ACL-intact patients, the most frequent type of tear was complex tears (60.5%, 98/162), but in ACL-injured patients, longitudinal tears were the most frequent ones (56.6%, 56/99). The posterior horn and extension to other compartments (more than two compartments) were the most common locations, regardless of ACL injury.

For lateral meniscal tears, the accuracy of MRI in patients with ACL injury was also found to be lower than that in ACL-intact patients (72.9% vs. 88.0%, *p* < 0.001). There were differences in the tear location (*p* < 0.001), but not in the type of tears between these patients (*p* = 0.291). In the ACL-intact patients, the most frequently involved types were the one with tears in more than two compartments (47.6%, 81/170) and mid-body tears (32.4%, 55/170), but in ACL-injured patients, the posterior horn was the most frequently involved compartment (52.5%, 53/101).

In brief, the overall accuracy of MRI in determining medial and lateral meniscal tears in patients with ACL injury was lower than that in patients with intact ACL. Moreover, although the accuracy of MRI in determining the specific location and type of lateral and medial meniscal tears in the ACL-intact or injured group was found to be similar, its accuracy in determining longitudinal tears or posterior horn tears of the medial meniscus in ACL-injured patients was lower than the other values (Table 2 and Table 3). The intra-observer reliability (ICC) of MRI ranged from 0.683 to 0.787 (*p* < 0.05), and the interobserver reliability ranged from 0.511 to 0.702 (*p* < 0.05).

## 4. Discussion

The most important finding of this study is that the overall diagnostic values of MRI for meniscus tears were acceptable with an accuracy of 85.8%, but the specific types and location of meniscus tears were difficult to determine using MRI regardless of ACL injury. Moreover, the overall diagnostic values of MRI in determining medial and lateral meniscal tears was lower in ACL-injured patients than in ACL-intact patients, but the specific location and type of tears were found to be similar. In the case of medial meniscus, the most frequently involved compartment was the posterior horn, including its extension (more than two compartments), regardless of ACL injury. Longitudinal tear of the medial meniscus was the most frequent type of tear in ACL-injured patients, but complex tear was the most frequent in ACL-intact patients. On the other hand, in the case of the lateral meniscus, complex and longitudinal tears were the most frequent type regardless of ACL injury. The posterior horn and its extension were the most frequent locations of tears in ACL-injured patients, but the mid-body and its extension were the most frequent locations of tears in ACL-intact patients.

The overall diagnostic values of MRI in this study were comparable to those reported in previous studies, indicating that it could be an acceptable diagnostic tool for meniscal tears [14,20,21,22]. However, the diagnostic values of MRI in determining the specific type and location of meniscal tears were found to be lower than the overall values regardless of ACL injury. In the study by Anderson et al., the interobserver reliability of MRI values in determining the specific type and location of meniscal tears were moderate to substantial, ranging from 0.46 to 0.72, even with arthroscopic findings. Dunn et al. [23] also reported moderate to substantial agreement between arthroscopy and MRI findings in classifying the type and location of meniscal tears (kappa: 0.61–0.63). In another previous study that assessed the correlation of meniscal tears arthroscopy and MRI findings using the ISAKOS classification, the kappa values showed moderate agreement [24]. The results of this study were similar to those of previous studies in terms of agreements and diagnostic values, including sensitivity, specificity, and accuracy [14,21,22,23,24]. Moreover, lower diagnostic values and agreements were found in patients with ACL injury than in patients without ACL injury. Thus, MRI could be used as an acceptable diagnostic tool to detect the presence of meniscus tear, but not for detailed classification of the type and location of meniscus tears using the present ISAKOS criteria regardless of ACL injury.

There were differences in the rate of type and location of meniscal tears between ACL-injured and ACL-intact patients. For the medial meniscus, longitudinal tear on the posterior horn, which was called as “ramp lesion”, was the most frequent tear in ACL-injured patients, while complex tear on the posterior horn was the most frequent tear in ACL-intact patients. Because of the rotational injury mechanism involved in acute ACL injury, known as the contrecoup mechanism, the longitudinal tear in the posterior horn of the medial meniscus might be induced along with the increased stress on the meniscocapsular junction [25,26,27]. Longitudinal tear or complex tear on the posterior horn was the most frequent lateral meniscus tear in patients with ACL injury. An axial loading with increased posterior sliding of the lateral femoral condyle, the so-called pivot shift mechanism, could induce the engagement of the lateral meniscus into the femoral condyle, similar to a bump against the posterior tibia [27,28,29,30]. Therefore, the tear on the posterior horn of the lateral meniscus was found more frequently in ACL-injured patients than in ACL-intact patients in this study. In brief, the type and location of meniscal tears between ACL-injured and ACL-intact patients were different as were the diagnostic values of MRI.

This study has several limitations. First, this was a retrospective study. Since incomplete data were excluded from this study, there might be a difference in the rates of types or locations of meniscal tears. Second, it is difficult to determine whether poor diagnostic values of specific type and location of meniscal tears can be attributed to the MRI process itself or the ISAKOS criteria. Because the overall diagnostic values of MRI were found to be good, similar to those in previous studies [20,21,22,31], the accuracy of the ISAKOS classification may be debatable. Third, in terms of specific type and location, relatively small numbers of meniscal tears were included. Furthermore, some types of meniscal tears, such as bucket handle tears or root tears, could not be distinguished in this study. Fourth, although the same surgical procedures were applied, some bias may have been introduced by the different surgeons in our institution. Furthermore, the diagnostic values might be different if the results of clinical examinations were analyzed simultaneously [32,33]. The reliability of arthroscopic findings was not assessed because arthroscopic findings were collected from arthroscopic surgical report form referring to arthroscopic images. However, all arthroscopic surgeries were performed by experienced surgeons of knee division. Therefore, we believe that reports on arthroscopic surgical report form are reliable. Fifth, there might be differences of diagnostic values according to the age, mechanism of meniscus tears or time from MRI to surgery, although it could not be evaluated in this study. This issue would be another good theme for another study about meniscus tear in ACL injury.

## 5. Conclusions

Preoperative MRI could be used as a diagnostic tool to identify for meniscus tears, but it is not capable of classifying the type and location of meniscus tears. Moreover, there were differences in the type and location of meniscus tears between ACL-injured and ACL-intact patients; hence, care should be taken during arthroscopic assessment of patients with ACL injury.

## Figures and Tables

**Figure 1 jcm-10-00606-f001:**
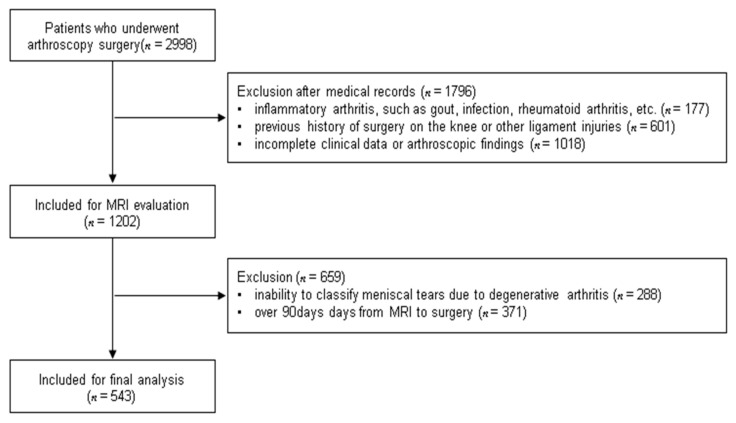
Flowchart of the included patients.

**Figure 2 jcm-10-00606-f002:**
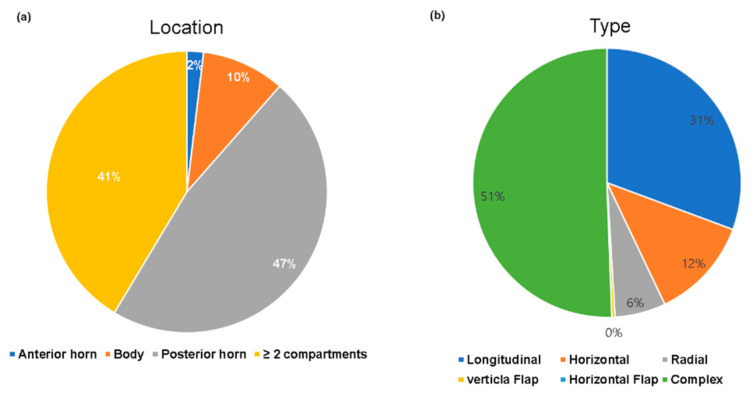
The distribution of medial meniscus tears on arthroscopy. (**a**) Location of medial meniscus tears. (**b**) Type of lateral meniscus tears. Complex tear was the most frequently observed type and the posterior horn was the most involved location.

**Figure 3 jcm-10-00606-f003:**
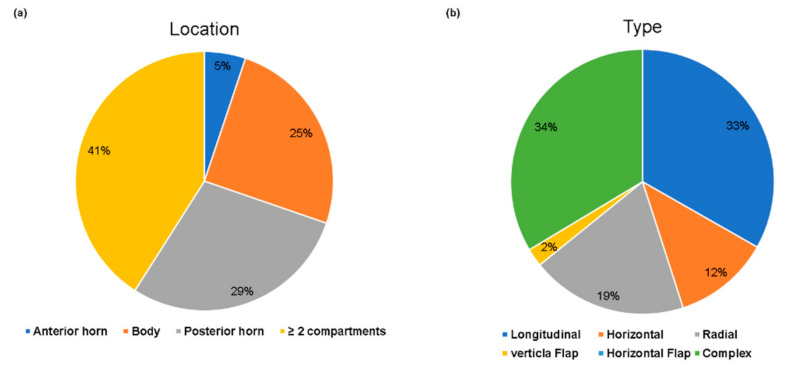
The distribution of lateral meniscus tears on arthroscopy. (**a**) Location of lateral meniscus tears. (**b**) Type of lateral meniscus tears. Complex tear and involvement of more than two compartments were the most commonly observed features.

**Table 1 jcm-10-00606-t001:** Summary of demographic data.

		Total Number of Patients	Anterior Cruciate Ligament (ACL) Injury	No ACL Injury	*p*-Value
		543	192	351	
Age		43.9 ± 17.4	37.1 ± 14.6	46.9 ± 17.5	<0.001
Sex					
Male		379	131	248	0.556
Female		164	61	103	
**Meniscus tear on arthroscopy**					
Medial meniscus	+	261	99	162	0.228
	−	282	93	189	
Lateral meniscus	+	271	101	170	0.353
	−	273	91	181	
**Meniscus tear in magnetic resonance imaging (MRI)**					
Medial meniscus	+	302	110	192	0.561
	−	241	82	159	
Lateral meniscus	+	258	86	172	0.347
	−	285	106	179	
Time from MRI to surgery (days)		22.1 ± 9.2	21.9 ± 7.3	23.3 ± 9.8	0.080

**Table 2 jcm-10-00606-t002:** Diagnostic accuracy of MRI for medial meniscus tear in ACL-injured knees.

	Sensitivity	Specificity	Accuracy	PPV	NPV
Type					
Longitudinal	48.2%	91.9%	79.2%	71.1%	81.2%
Horizontal	33.3%	96.2%	94.3%	22.2%	97.8%
Radial	66.7%	92.1%	91.7%	11.8%	99.4%
Vertical flap	0.0%	99.5%	99.5%	0.0%	100%
Horizontal flap	0.0%	99.5%	99.5%	0.0%	100%
Complex	76.5%	88.6%	86.5%	59.1%	94.6%
Location					
Anterior horn	0.0%	98.4%	97.4%	0.0%	98.9%
Body	50.0%	100.0%	98.4%	100.0%	98.4%
Posterior horn	64.8%	83.3%	78.1%	60.3%	85.8%
More than two compartment	67.6%	85.8%	82.3%	53.2%	91.7%

Data given as percentage (%).

**Table 3 jcm-10-00606-t003:** Diagnostic accuracy of MRI for lateral meniscus tear in ACL-injured knees.

	Sensitivity	Specificity	Accuracy	PPV	NPV
Type					
Longitudinal	31.4%	96.2%	84.4%	64.7%	86.3%
Horizontal	14.3%	94.6%	91.7%	9.1%	96.7%
Radial	58.3%	95.2%	90.6%	63.6%	94.1%
Vertical flap	0.0%	98.4%	97.4%	0.0%	98.9%
Horizontal flap	0.0%	99.5%	99.5%	0.0%	100%
Complex	89.9%	88.6%	82.8%	50.0%	89.4%
Location					
Anterior horn	80.0%	96.8%	96.4%	40.0%	99.5%
Body	30.8%	97.2%	92.7%	44.4%	95.1%
Posterior horn	43.4%	89.2%	76.6%	60.5%	80.5%
More than two compartment	50.0%	91.9%	85.4%	53.6%	90.9%

Data given as percentage (%).

**Table 4 jcm-10-00606-t004:** Diagnostic accuracy of MRI for medial meniscus tear in ACL-intact knees.

	Sensitivity	Specificity	Accuracy	PPV	NPV
Type					
Longitudinal	58.3%	98.2%	95.4%	70.0%	96.9%
Horizontal	69.2%	86.8%	85.5%	29.5%	97.2%
Radial	38.5%	94.9%	92.9%	22.7%	97.6%
Vertical flap	0.0%	100%	99.7%	0.0%	99.7%
Horizontal flap	0.0%	99.7%	99.7%	0.0%	100%
Complex	65.3%	90.5%	83.5%	72.7%	87.1%
Location					
Anterior horn	0.0%	99.2%	98.3%	0.0%	99.1%
Body	21.1%	97.9%	93.7%	36.4%	95.6%
Posterior horn	62.3%	87.6%	82.6%	55.1%	90.5%
More than two compartment	85.9%	85.7%	85.8%	60.4%	96.0%

Data given as percentage (%).

**Table 5 jcm-10-00606-t005:** Diagnostic accuracy of MRI for lateral meniscus tear in ACL-intact knees.

	Sensitivity	Specificity	Accuracy	PPV	NPV
Type					
Longitudinal	61.8%	96.6%	91.2%	77.3%	93.2%
Horizontal	48.0%	94.2%	90.9%	38.7%	95.9%
Radial	53.6%	91.0%	88.0%	34.1%	95.8%
Vertical flap	75.0%	99.7%	99.4%	75.0%	99.7%
Horizontal flap	0.0%	100%	100%	0.0%	100%
Complex	44.8%	91.1%	83.5%	50.0%	89.3%
Location					
Anterior horn	55.6%	94.7%	93.7%	21.7%	98.8%
Body	47.3%	95.3%	87.7%	65.0%	90.7%
Posterior horn	48.0%	97.5%	94.0%	60.0%	96.1%
More than two compartments	72.8%	91.9%	87.5%	72.8%	91.9%

Data given as percentage (%).

## Data Availability

The data presented in this study are available on request from the corresponding author. The data are not publicly available due to Personal Information Protection.

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
