# Peer review of "Diagnostic Accuracy of Magnetic Resonance Imaging in the Detection of Type and Location of Meniscus Tears: Comparison with Arthroscopic Findings"

_jcm, 2021, doi:10.3390/jcm10040606_

Round 1

Reviewer 1 Report

Dear authors,

I had the pleasure to review your manuscript. Please find my comments hereafter.

Abstract:  

  • Since you exclude almost 5 out of 6 patients, you should give some exclusion/inclusion criteria to your abstract.
  • You should include the type of MRI used (1.5 or 3T).
  • Give some information to the time frame the patients/images were included in this study

Introduction

  • adequate

Methods

  • 1: how did you classify root tears ?
  • 2 it is very astonishing that you performed trans-notch visualisation for all knees as early as in 2005 (especially since your references are from 2018 and 2020). Please confirm your methodology that it was identical from 2005 to 2018.
  • Who performed the measurements ? who performed the inter- and intra-rater reliability? What experience ?
  • Who performed the arthroscopy ? did you re-evaluate the arthroscopic pictures or did you take the information from the surgical report form ? Since you mention more than 1000 drop-outs due to incomplete clinical data (which one ? you do not include clinical data), or arthroscopic findings, “systematic” arthroscopic diagnostic evaluations is very relative, since you include only a little bit more than 543 patients.
  • Especially ramp lesions are not seen on MRI (as mentioned in your introduction), however there are indirect signs of ramp lesions, similar to incomplete radial tears or instabilities with minimal extrusion of the meniscus. I therefore think, your methodology needs to be described more into details
  • Which horizontal tears did you include from your MRI evaluations ? also grade II ? or only grade III ? please be more precise for all types of meniscal lesions.

Results

  • Your results are very difficult to read. I suggest to shorten the text, and to point out the most important findings from your tables.
  • Did you perform also an inter-rater reliability for reading arthroscopic imaging ?

Discussion

  • Your primary outcome was to compare arthroscopy vs MRI, and your secondary outcome was to make a subgroup analysis. Please try to structure your discussion also this way, by starting by your primary, then your secondary outcome (this is also true for your abstract and conclusion).

Conclusions

  • adequate

Figures

  • Figure 1 and 2: please give the information in the legends, whether these are arthroscopic or MR findings.

Tables

  • adequate

References

  • adequate

Author Response

Thank you for your comments. 

Reviewer 2 Report

It was a pleasure to read your well written original article on a retrospective analysis of the diagnostic accuracy of magnetic resonance imaging in the detection of type and location of meniscus tears, comparing with arthroscopic findings.

2998 patients were eligible and finally 543 patients were included for 14 years and MRI findings were correlated with intraoperative arthroscopic findings. Inter- and intraobserver reliability were calculated. Groups with and without ACL injuries were compared for accuracy of MRI. This adds to the novelty of the current study, as to my knowledge a differentiation between MRI accuracy of meniscus tears in ACL injured knees and knees without ACL injury does not exist yet.  There are major and minor considerations listed below that should be integrated in your manuscript to make it suitable for publication in Journal of Clinical Medicine. I recommend that a major revision is warranted.

Major comments:

  • Please include a Flowchart of patient eligible and included in the study for better illustration. Also list the exclusion criteria in the chart and the number of patients that received arthroscopy.
  • Please explain in the text, how it can be explained that a great number of patients (1018) had to be excluded due to incomplete documentation? How could the surgical reports be incomplete in so many cases?
  • It would be interesting to assess a correlation of your findings regarding the MRI diagnostics with clinical findings e.g., meniscal test (McMurray test, the Apley test, the Stienmann I test, the Payr’s test, Childress’ sign, and the Ege’s test). Did you assess according data? If not please discuss based on literature, for example Ercin et al. 2012, Knee Surgery, Sports Traumatology, Arthroscopy volume or Antinolfi et al. 2018, Joints.
  • Please describe ACL injuries more exactly: Where all injuries fully ruptured ACLs? Or were incomplete ruptures also seen?
  • Include the patient´s age in the demographic data as different location and morphologies of meniscal tears are most likely to be associated with great differences in patient age. Please also provide information about the trauma mechanisms leading to meniscal and ACL injuries.
  • How many surgeons performed surgery on patients included in the study?
  • Table 1 last line: Time from MRI to surgery. Please briefly discuss, if you expect an influence in the arthroscopic diagnosis based on the time between MRI to surgery. 

Minor comments:

  • Line 8: Please double-check grammar.
  • Lines 242 to 248: Sentence “Thus, MRI (…) ACL injury” is double.

Author Response

Thank you for your comments. 

Round 2

Reviewer 1 Report

Dear authors,

The paper has been changed accordingly to most of the reviewer's comments, and otherwise were answered sufficiently. 

I have no more comments about your paper. 

The result section could be still more concise (still very long).

Reviewer 2 Report

Thank you for including the proposed changes into your manuscript. I believe the article gained of quality and see it warranted for publication in JCM. Congratulation.